# Congenital syphilis in Argentina: Experience in a pediatric hospital

**Luciana Noemí García**[1,2☯]\*, **Alejandra Destito Solján**[1☯], **Samanta Moroni**[1],
**Nicolas Falk**[1,2], **Nicolás Gonzalez**[1], **Guillermo Moscatelli**[1,2], **Griselda Ballering**[1],
**Facundo García Bournissen**[1,2], **Jaime M. Altcheh**[1,2]

**1** Servicio Parasitología- Chagas, Hospital de Niños Ricardo Gutierrez, Capital Federal, Buenos Aires, Argentina, **2** Instituto Multidisciplinario de Investigaciones en Patologías Pediátricas (IMIPP), CONICET, Buenos Aires, Argentina

☯ These authors contributed equally to this work.
\* binlugarcia@gmail.com

**Data Availability Statement:** All relevant data are within the manuscript and its Supporting Information files.

## Abstract

In spite of being preventable, Congenital syphilis (CS) is still an important, and growing health problem worldwide. Fetal infection can be particularly aggressive, but newborns can be asymptomatic at birth and, if left untreated, develop systemic compromise afterwards with poor prognosis. We analyzed 61 CS diagnosis cases between 1987–2019 presenting at the Buenos Aires Children' Hospital. The distribution of cases showed a bimodal curve, with a peak in 1992–1993 and in 2014–2017. Median age at diagnosis was 2 months (IQ 1–6 months). The main clinical findings were: bone alterations (59%); hepatosplenomegaly (54.1%); anemia (62.8%); skin lesions (42.6%) and renal compromise (33.3%). Cerebrospinal fluid (CSF) was abnormal in 5 patients, normal in 45 and was not available for 11 patients. Remarkably, spinal lumbar puncture did not modify therapeutic decisions in any case. Between mothers, only 46% have been tested for syphilis during pregnancy and 60.5% patients had non-treponemal titers equal to or less than fourfold the maternal titer. Intravenous penicillin G was prescribed for all except one patient, who received ceftriaxone with good therapeutic response. During follow-up, 1.6% infants died, 6.5% had persistent kidney disorders and 1.6% showed bone sequelae damage. RPR titers decreased after treatment, reaching negative seroconversion in 43% subjects at a median of 26.4 months. Low adherence to follow up was observed due to inherent vulnerable and low-income population characteristics in our cohort. Our results highlight a rising tendency in cases referred for CS in our population with high morbidity related to delayed diagnosis. A good therapeutic response was observed. CS requires a greater effort from the health system to adequately screen for this disease during pregnancy, and to detect cases earlier, to provide an adequate diagnosis and treatment.

## Author summary

Congenital syphilis (CS) is caused by mother-to child transmission in a setting with poor maternal screening strategies for the infection. Although screening of pregnant women

**Funding:** This work was funded by financial support of the Hospital de Niños Ricardo Gutierrez (http://guti.gob.ar/). The funder has no role in study design, data collection and analysis, decision to publish, or preparation of the manuscript.

**Competing interests:** The authors have declared that no competing interests exist

and treatments are simple, inexpensive and widely available, new cases are increasing worldwide. We reviewed the medical records of CS-patients assisted in our hospital over the past 30 years. Our results showed an increase in the number of CS cases. At birth, most children were asymptomatic and later developed CS clinical manifestations. Penicillin treatment, and in one case ceftriaxone, was prescribed with a good clinical and serological response. Nevertheless, one infant died, four had persistent kidney involvement and one showed bone sequelae. Spinal lumbar puncture did not modify therapeutic decisions. We conclude that the detection and treatment of CS, as well as consistent follow-up, remains a great challenge for clinical practice in our region, particularly in underserved patients. Also, we observed that the commonly used diagnostic lumbar puncture had no impact in the management of asymptomatic newborns, and its role in CS should probably be reevaluated.

It is crucial that the health system, and pediatricians and obstetricians in particular, make a greater effort to detect this neglected disease in an attempt to reverse its upwards trend.

## Introduction

Congenital syphilis (CS) is the result of the transplacental infection by *Treponema pallidum pallidum* (TPA) and mainly affects socioeconomically disadvantaged vulnerable populations [1]. In 2015, the Pan American Health Organization (PAHO) estimated that there were at least 22,800 cases of CS (ie: 1.7 cases per 1,000 live births) in Latin America [2]. In Argentina syphilis cases have more than doubled in recent years mainly as a result of an increase in primary syphilis in the population of childbearing age, which has produced an increase in the number of CS cases [3]. In this scenario, adequate screening and treatment of infected mothers is vital to avoid CS [1,4].

Another window of opportunity for the diagnosis of CS is the neonatal and infancy period. Most infected newborns are asymptomatic and, if left undiagnosed and untreated, the infection persists silently. As a result of this, inflammatory response continues in tissues and may lead to CS manifestations months or years later [5]. Initial signs, such as prematurity, hematological and dermatological manifestation, pneumonia, nephrotic syndrome (among others) occur before the age of 2 years and are denominated early CS; while individuals presenting with signs and symptoms after the age of 2 years have predominantly affect the central nervous system, bones, joints and teeth, with the dominated late CS [6].

Our main objective was to describe the medical experience relating to the detection, treatment, clinical and serological evolution of patients with CS assisted in our hospital during the last 30 years.

## Methods

### Ethics statement

The Ethics Committee and Review Board (named: Comité de Ética en Investigación) of the Hospital de Niños Ricardo Gutierrez approved this study (approval number: DI-2020-159-GCBA-HGNRG). The study is registered in clinical trials.gov (NCT04137601) and adhered to the tenets of the Declaration of Helsinki. Written consent was obtained from the parent/guardian of all patients to use their data and/or images for academic purposes anonymously.

We conducted a prospective cohort study with retrospective data collection of pediatric patients with congenital syphilis assisted at the Servicio de Parasitología-Chagas, Hospital de Niños "Ricardo Gutierrez", a tertiary care hospital without a maternity unit, between February 1987 and June 2019.

CS cases were defined according to Argentina National Guidelines [7] which are homologous to the global surveillance case definition [8] and the scenarios 2 (Possible Congenital Syphilis) and 1 (Proven or highly probable congenital syphilis) described by the Centers for Disease Control and Prevention for the congenital syphilis evaluation and treatment [1].

The CS case definition was as follows:

- A newborn or infant with a positive non-treponemal test (VDRL or RPR), born to a mother who had untreated or inadequately treated syphilis; and clinical evidence of congenital syphilis or pathologic long-bone X-rays; or Cerebrospinal fluid (CSF) alteration and positive non-treponemal test; or increase in CSF protein with no alternative causes.

- Exclusion criteria: patients with acquired syphilis.

According to national guidelines [7] adequate syphilis screening during pregnancy was defined by at least one nontreponemal serological test at the first trimester and third trimester. In the same way, adequate mother treatment was considered when it was carried out with penicillin, (3 doses separated by 1 week each one and being the last dose administered one month before delivery) [7].

Demographics, clinical findings, TPHA/RPR serology, general laboratory, complementary studies, and treatment prescribed data were collected.

The serological tests used were the nontreponemal test: Rapid plasma reagin (RPR) or Venereal Disease Research Laboratory (VDRL) and the treponemal test: *Treponema pallidum* haemagglutination (TPHA). For serological follow-up analysis, we included patients with at least two RPR tests, done at the initial visit and repeated between 2 to 6 six months after treatment (see S1 Fig).

Maternal data were collected for serological pregnancy control and antibiotic treatment.

Data were extracted from clinical charts and anonymized by issuing a random-number' identifier to each individual patient. The master list linking patients identifying information (i.e. name, date of birth and chart number) to the study identifier was kept in a secure location to which only the senior and principal investigators had access.

Descriptive statistics were used for the variables of interest. Continuous variables are presented as means with 95% CI or medians and interquartile range. Categorical variables are represented in percentages. The disappearance kinetics of serum antibodies were analyzed using survival analysis (Kaplan-Meyer). Analyses were performed with R software v3.0 (R Core Team 2018. R Foundation for Statistical Computing, Vienna, Austria. https://www.R-project.org/).

## Results

Out of 100 clinical charts analyzed, 61 patients fulfilled inclusion criteria for CS. The remaining 39 were patients with acquired (i.e. non-congenital) syphilis. The data of complementary studies is summarized in a flowchart in S1 Fig.

The incidence of CS cases showed a bimodal time curve with a peak in 1992–1993 and another in 2014–2017 (Fig 1).

CS patients' mothers were in general young with a median age of 24 (IQ$_{25-75}$: 19–36). Adequate TPA serological control [7] was carried out in only 27/57 (47%) women during pregnancy. Also, 27/57 (47%) were not properly screened and data was missing for 4/61 (6.5%) pregnancies (3 children were adopted and one mother died postpartum). Only 4/57 (7%)

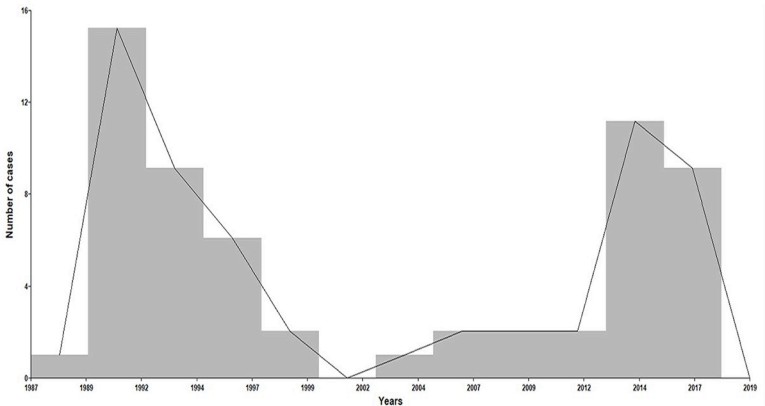

**Fig 1. Number of congenital syphilis cases.** Histogram shows registered cases by year.

mothers had received penicillin treatment before delivery. Three mothers received 3 doses of benzathine penicillin in the second trimester and the fourth mother received two regimens of 3 doses of penicillin ending within the last month of pregnancy.

Out of the 61 CS patients in the cohort, 15/61 (24.6%) were born preterm (<37 weeks), 15/ 61 (24%) were born full term but with low birth weight or Intrauterine Growth Restriction (IUGR) and 31/61 (50.8%) were full term babies. Median age at diagnosis was 2 months (IQ$_{25-75}$: 1–6 months), ranging from 1 day to 8.5 years. Most patients were diagnosed during the period of early CS, with an accumulation of cases between one to six months of age, while 6 children exhibited late CS and were diagnosed older than 2 years (Fig 2). In 3 patients congenital coinfection was detected: cytomegalovirus in 2 patients and hepatitis B in 1. All patients were HIV negative.

At birth, 48/61 (78%) were asymptomatic. In the following weeks 40/48 developed symptoms of CS (median: 2, IQ$_{25-75}$: 2–4 months) and 8 remained asymptomatic. These asymptomatic cases were diagnosed at different ages (median: 21, IQ$_{25-75}$: 0–24 months) as follows: 2 cases were siblings of an index case, 2 cases by pre-surgical serological testing and 4 by serological screening during the follow-up because their mothers tested positive for syphilis. In summary, at diagnosis, 53 (86.8%) CS infants were symptomatic (S2 Fig) and 8 (13.1%) asymptomatic

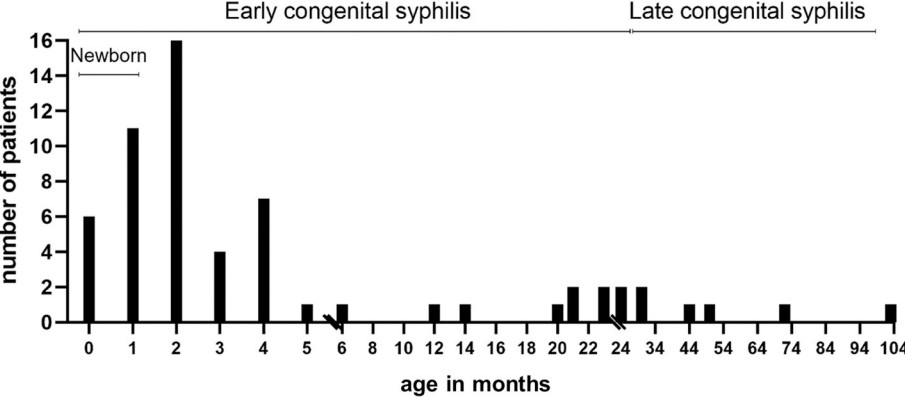

**Fig 2. Timeline of the age at diagnosis.** Bar chart shows the number of patients diagnosed at different ages and the period of early and late congenital syphilis.

Demographic data showed that only 11 (18%) patients lived in Buenos Aires City, while 41 (67.3%) did so in nearby cities and 3 (5%) came from neighboring countries. In 6 (9.8%) cases data was not available. Low socioeconomic status and limited access to the health system were common features for all patients and their families.

As regards clinical findings (Table 1), most patients, 36/61(59%), showed bone involvement (Fig 3) and all of them express periostalgia as confirmed by bone X-ray. Among patients (13/36) with Parrot's pseudoparalysis the right upper limb was the most frequently affected. Over half of patients presented with hepatosplenomegaly (54.1%) as well as dermatological manifestations (50.8%) (Fig 4).

Complementary studies at diagnosis are presented in Table 2. Alterations in hepatic parameters were observed in 17/47 (36%) of the patients. Data was missing for the remaining 14 patients. Severe anemia, with hemoglobin lower than 7.5 g/l and hemodynamic compromise, which required red blood cell transfusion, was observed in 5/61 (8.2%). Ocular fundus examination showed alterations in 4/36 (11%) patients evaluated by fundoscopy, while abnormal hearing evaluations was only detected in 1/19 (5.3%) patients with hearing testing. Lumbar puncture was performed in 50/61 patients (82%): CSF was altered in 5/50 (10%) patients: 4 showed reactive VDRL and 1 showed a high white blood cell count. In 9/50 patients (18%) lumbar puncture was traumatic. Cranial ultrasound was normal in 23/23 patients studied. Not all data was available for all patients.

TPA serology at diagnosis: RPR/VDRL showed median titers of 32 dilution (IQ$_{25-75}$: 4–128), and TPHA showed median titers of 640 dilution (IQ$_{25-75}$: 160–2560).

A total of 38 paired mother and child samples were available for RPR/VDRL titers comparison. A considerable number (39.4%) of patients did not reach 4-fold titers more than their mothers and in 28.9% the titers were the same or even lower (Table 3).

**Table 1. Demographic and clinical findings at diagnosis.**

| Age | | |
|---|---|---|
| Median | 2 months | |
| Range | 1 day- 8.5 years | |
| IQ $_{25-75\%}$ | 1–6 months | |
| **Sex** | **Relative frequency** | **%** |
| Female | 29/61 | 47.5 |
| Male | 32/61 | 52.5 |
| **Clinical findings** [a] | **Relative frequency** | **%** |
| Bone involvement | 36/61 | 59 |
| Periostalgia | 36/36 | 100 |
| Pseudoparalysis | 13/36 | 36 |
| Saddle nose | 1/36 | 2.7 |
| Saber shins | 2/36 | 5.5 |
| Hepatosplenomegaly | 33/61 | 54.1 |
| Dermatological manifestations | 31/61 | 50.8 |
| Maculopapular rash | 17/31 | 54.8 |
| Palms and soles' rash | 17/31 | 54.8 |
| Pemphigus | 15/31 | 48.4 |
| Desquamation lesions | 11/31 | 35.4 |
| Jaundice | 11/31 | 35.4 |
| Condyloma lata | 2/31 | 6.4 |
| Rhinitis | 10/61 | 16 |
| Fever | 15/61 | 24.6 |
| Hutchinson's teeth | 1/61 | 1.65 |

[a] some patients developed more than one clinical feature or had several laboratory findings

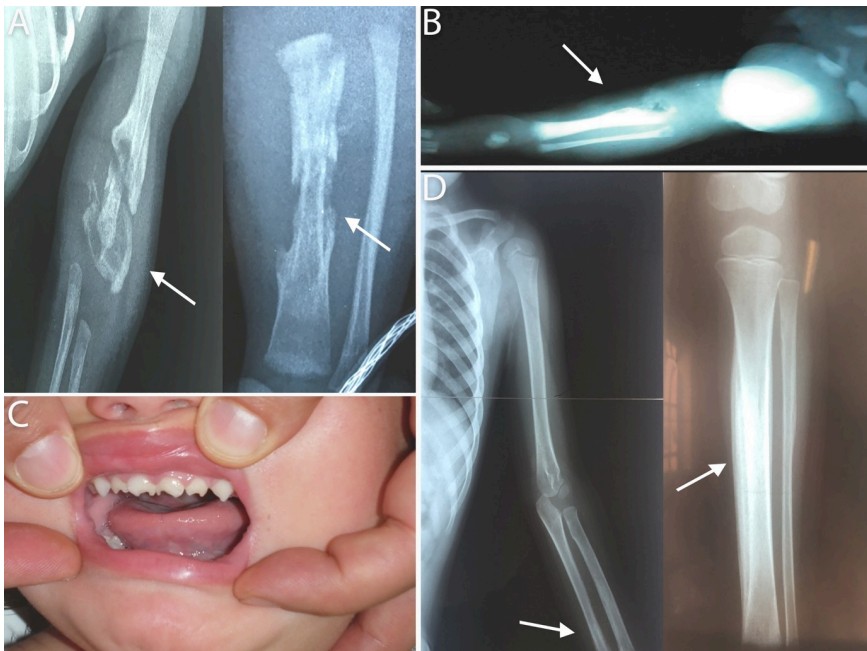

**Fig 3. Representative radiological images of bone involvement and Hutchinson's teeth findings.** (A) Posteroanterior radiographs of a female newborn showing, at the left, humerus with diffuse periosteal reaction (arrow) and, in the right, lytic image in the distal region of left tibia (arrow). (B) Posteroanterior radiograph of a male newborn showing lytic image in the proximal region of the right tibia and fibula. (C) Hutchinson's teeth in a 2- years old girl (D) Posteroanterior radiographs showing widespread periostitis of the humerus (arrow), at the left, and tibia at the right (arrow) in a 2- years old girl.

One patient (1/61) had a false-negative RPR test at birth but eventually developed symptoms of late CS (Hutchinson's teeth) and RPR was found to be positive (Fig 3C).

Almost all CS patients (60/61) were treated with intravenous aqueous crystalline penicillin G for 10–14 days (median: 12, 95% CI:10–14). Doses were age- and weight-adjusted and covering possible involvement of central nervous system (CNS), following national guidelines [7]. Jarisch-Herxheimer reaction (fever and vasomotor signs) occurred in 9/61 (14.7%) CS patients, with onset within 12 hours after beginning treatment and a maximum duration of 36 hours. One patient started treatment, because of suspected sepsis, with ceftriaxone 50 mg/kg/day for ten days. Overall 47/53 (89%) of symptomatic CS patients completely recovered after treatment (Figs 4B and 5B), and 4 patients had residual organ involvement.

In 4 patients persistent proteinuria was observed as a sequelae of renal involvement. In relation to this, two patients (5 and 4 months old) developed nephrotic syndrome secondary to CS, with massive proteinuria and subsequent complete recovery after 4 months of penicillin treatment. The other two patients (45 days and 3 months old) developed nephrotic syndrome secondary to CS, with mild proteinuria and hematuria that also resolved 3 months after treatment in the 45-day-old infant, but unfortunately the 3-month-old died of sepsis due to Escherichia Coli, severe anemia and renal failure. In all cases, other causes for the renal symptoms were ruled out without reaching biopsy due to the subsequent recovery. One patient (3 years old) developed elbow and knee arthrosis secondary to CS bone damage (Fig 6). The patient was symptomatic at birth with liver involvement (cholestasis), neurological (pathological CSF), skin and osteoarticular compromise. After several surgeries (varising osteotomies with non-vascularized fibula autograft in tibia and humerus) and subsequent compression osteosynthesis, the patient is walking again and under follow-up.

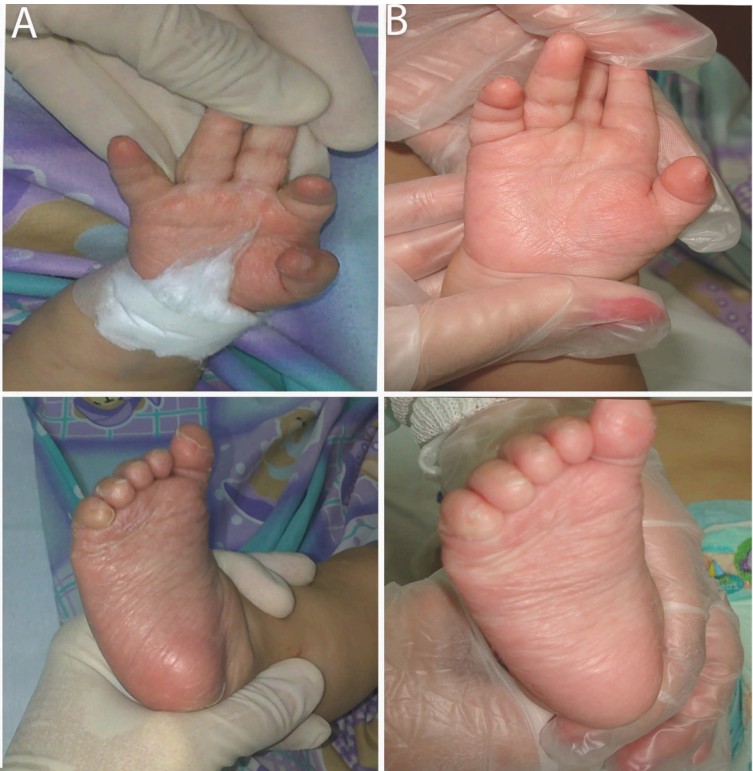

**Fig 4. Dermatological findings.** (A) Pictures demonstrating maculopapular rash on the palms (upper panel) and soles (lower panel) of a 4-month-old boy. (B) Improvement of the lesions two days after penicillin treatment in the same patient.

Serological follow-up: data was available for 30/61 patients. For those patients, the median follow-up time was 12.1 months (RIQ: 4.1–25.6 months). A decrease of VDRL/RPR titers was observed, (Fig 7), becoming negative in 13/30 (43%) patients at a median 26,4 months (95%CI: 14.2-infinity) after treatment. A decrease in TPHA titers was also observed, but only 2 patients showed serological negativization at 5 years of follow-up.

## Discussion

Over the last 30 years, our service as a pediatric referral center, assisted over 61 patients suffering from CS, which is a potentially eradicable disease. In Argentina an increase in primary syphilis was reported over the last few years [3]. In order to see whether the rising trend of primary syphilis had an impact on the occurrence of new cases referred for CS, we conducted a retrospective analysis of a cohort of assisted CS patients in our hospital.

In our study, we provided cumulative data about the number of cases, clinical presentation, squeals and follow-up serology of CS attended in a third level hospital without maternal ward. The tendency of our data is similar to worldwide trends and the scarce available data in LAC and Argentina in the last decades. Vacancy of antenatal care visits and newborn follow-up, as well as the health worker training on clinical suspicion of syphilis observed in our cohort reinforce the difficulties reported by Argentina and the LAC countries [9].

Our results showed an increase in the number of CS cases referred to our center in recent years. Interestingly a bimodal curve with a peak of cases in the early 90's was observed and another peak in 2014–2017. A similar trend has been highlighted in several countries

**Table 2. Complementary studies at diagnosis.**

| Routine laboratory data and complementary studies [a] | Relative frequency | % |
|---|---|---|
| Urinalysis | 45/61 | 73.7 |
| Normal | 30/45 | 66.6 |
| Altered | 15/45 | 33.3 |
| Mild proteinuria | 8/45 | 17.7 |
| Massive proteinuria | 4/45 | 8.9 |
| Hematuria | 9/45 | 20 |
| Leukocyturia | 2/45 | 4.4 |
| Liver functions | 47/61 | 77 |
| Normal | 30/47 | 63.8 |
| Altered | 17/47 | 36.2 |
| Hyperbilirubinemia. | 14/47 | 29.7 |
| Alanine transaminase elevation | 12/47 | 25.5 |
| Complete Blood Count | 51/61 | 83.6 |
| Anemia | 30/51 | 58.8 |
| Leukocytosis | 16/51 | 31.4 |
| Thrombocytopenia | 5/51 | 9.8 |
| Ocular fundus examination | 36/61 | 59 |
| Normal | 32/36 | 89 |
| Altered | 4/36 | 11 |
| Keratitis | 1/36 | 2.8 |
| Retinal pigmentation | 2/36 | 5.5 |
| Fovea alterations | 1/36 | 2.8 |
| Bone Radiological Findings | 61/61 | 100 |
| Normal | 22/61 | 36 |
| Altered | 36/61 | 59 |
| Periostitis | 36/61 | 59 |
| Osteochondritis/osteomyelitis | 6/61 | 9.8 |
| Fracture | 3/61 | 5 |
| Osteolytic lesions | 2/61 | 3.2 |
| Missed diagnosis | 3/61 | 5 |

[a] some patients developed more than one clinical feature or had several laboratory findings

worldwide [5,10]. In developed countries of America, the first increase was attributed to the new diagnostic definition of CS in 1988 [5], and the second peak was due to an increase in reported cases of primary syphilis in women from 2015 onwards [10,11].

The lack of high-quality data about maternal and congenital syphilis in Latin America Countries (LAC), which has been reported by health organizations on several occasions [9,12,13], makes correlating our findings with the local and global epidemiological trends difficult. Maternal and congenital syphilis declined worldwide between 2008 and 2012, since the Strategy and Plan of Action for the Elimination of Mother-to-Child Transmission (EMTCT) was launched, which seems to be reflected in our data as a dip in the number of CS cases [9]. However, LAC continued to have the third-highest prevalence of maternal syphilis in the world in 2012, with CS rates from 1 in Honduras to 108 per 1000 in Panama [13]. Soon after, the number of cases increased in our service. This is coincident with the growth in syphilis in Argentina from 10.4 in 2012 to 51.1 per 100,000 in 2018 [3]. Congenital syphilis behaved similarly to other syphilis-related events, showing accelerated increased tendency from 1 per 1000 in 2013 to 1.3 in 2014 reaching 1.7 in 2017 and a decrease to 1.5 in 2018 [3]. This is reflected in our cohort as a peak in 2014–2017.

**Table 3. Comparative nontreponemal testing (Child vs mother).**

| RPR determination/ Matching | 4-fold | 2-fold | Equall | Lower |
|---|---|---|---|---|
| Child *vs* mother's titers | 23 (60.5%) | 4 (10.5%) | 3 (7.9%) | 8 (21%) |

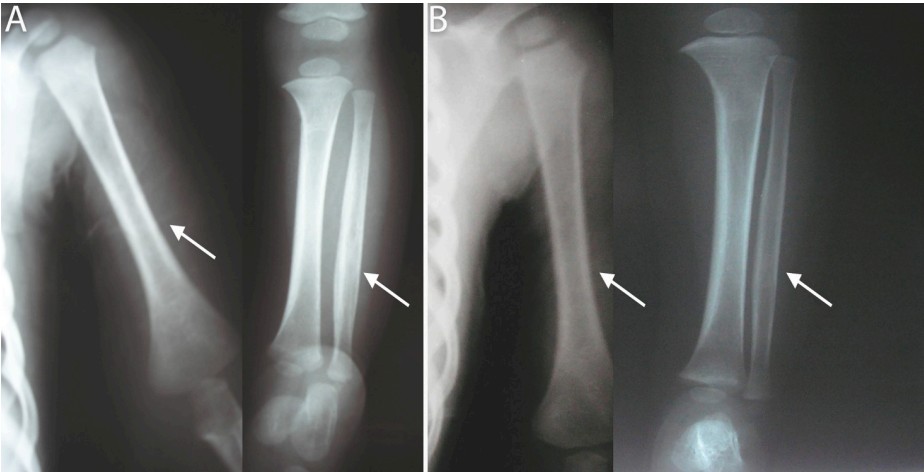

**Fig 5. Representative radiological images of the bone lesions evolution after treatment.** (A) Posteroanterior radiographs of a 14 month-old girl showing, at the left, diffuse periosteal reaction in the humerus (arrow) and in the right, periosteal reaction in tibia and fibula (arrow) (B) Improvement of the lesions after treatment.

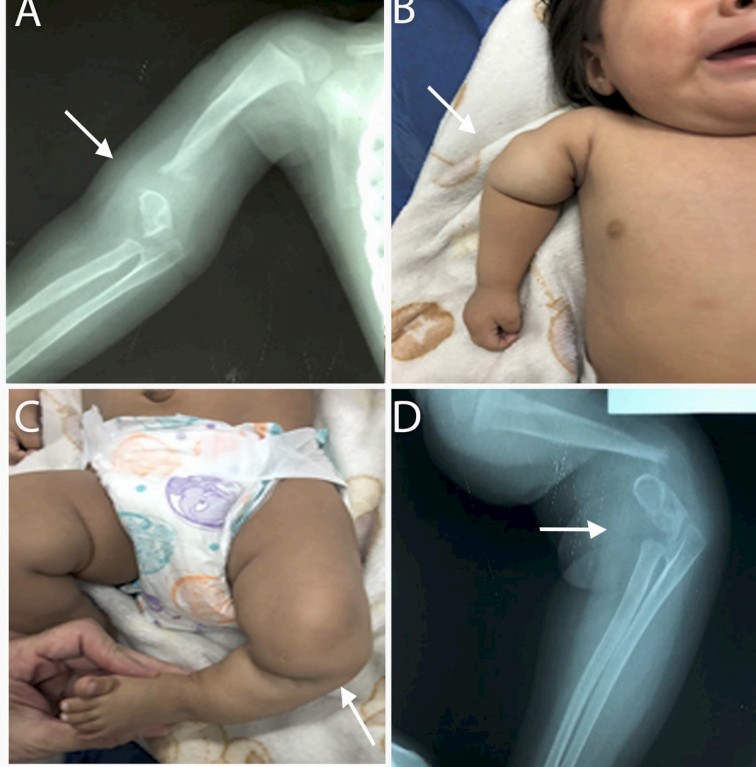

**Fig 6. Bone sequelae: Radiological images and pictures of one patient.** (A) Posteroanterior radiograph taken of a 3 years-old girl showing pseudarthrosis of the right elbow (arrow) with absence of distal metaphyseal ossification of the humerus and radio-ulnar osteoporosis. (B) Bone deformities (arrow) involving the junction of the right arm (C) Bone deformities (arrow) involving the junction of the left leg with severe anterolateral bowing of the limb (D) Lateral view radiograph showing left knee pseudoarthrosis (arrow) with lack of ossification of the proximal and distal metaphysis of the femur and distal tibia and fibula osteoporosis.

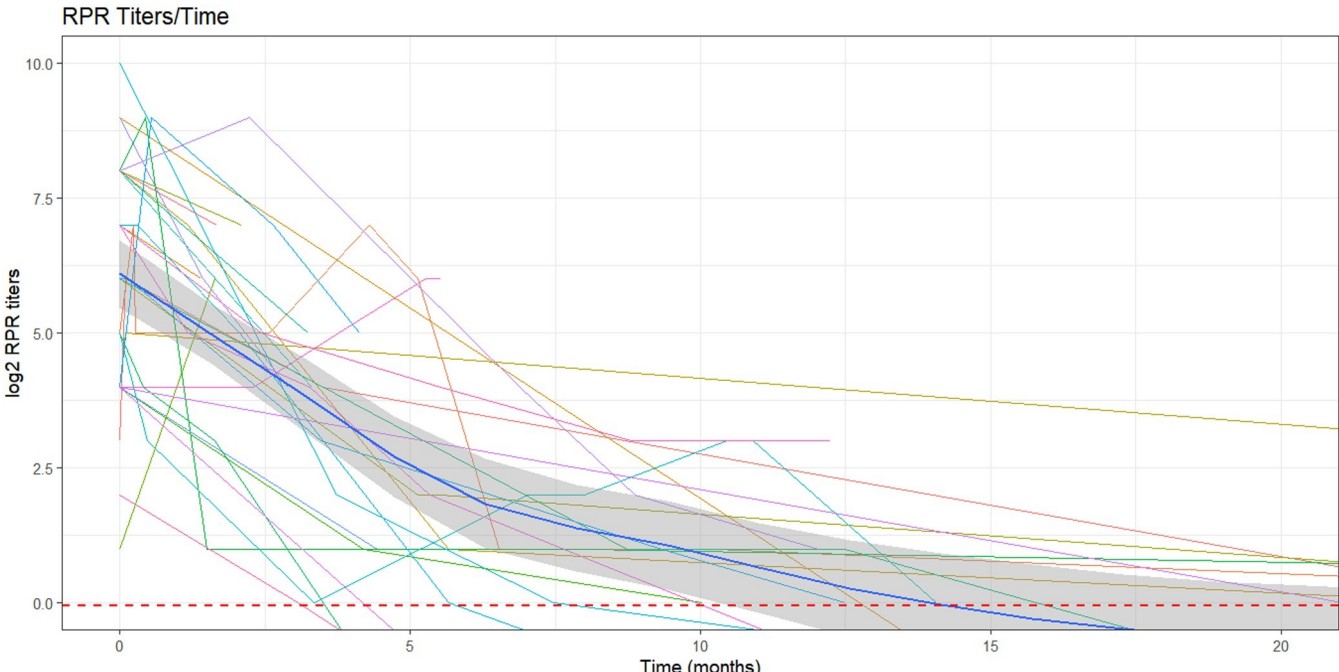

**Fig 7. Serological follow-up in 30 congenital syphilis treated patients.** Figure shows log titers of RPR changes over time. Dark blue line represents the smooth regression of the data. Red horizontal line represents the cut value.

Among possible causes of this trend, we observed that, despite the fact that 96% of CS patients in our cohort were born in medical centers in Argentina, a high percentage of women were not adequately screened for syphilis during pregnancy [3,12,14]. This led to missed opportunities to prevent syphilis transmission to the fetus [1,15]. In our study, only 47% of pregnant women had completed adequate prenatal serological screening, and only four mothers were treated appropriately (unfortunately, we do not have sufficient data to rule out a therapeutic failure for these 4 mothers).

The low percentage of screened pregnancies in our cohort is similar to the 40% rate of prenatal screening performance for syphilis reported in Argentina [3,12,14]. As part of the reality of our country, a great percentage arrive as untreated early syphilis at delivery and among them, perinatal screening is performed nearly in a 10.7% of the children [3,16]. On this subject, a survey performed to 80 obstetricians reported shortage of scheduled shifts, the unavailability of adequate medical records and the lack of formal referral circuits between hospitals and health centers as important issues that make screening and treatment of syphilis in pregnancy difficult [16].

We observed that 44.7% of our cohort lost systemic opportunities to prevent early infection for TPA; which supports the efforts pointed to serial serological tests among all the pregnancy and the perinatal study of the infants [17,18]. Additionally, newborns whose serologic tests for syphilis are reactive and whose mothers were treated for syphilis should undergo a comprehensive evaluation consisting of a detailed physical examination and close follow-up at three/four and six months of age until VDRL/RPR reactivity declines or disappears [19].

Globally, maternal syphilis is the most common infection associated with stillbirth in low-resource settings [20]. It is estimated that nearly 20% of CS newborns from untreated mothers, express IUGR prematurity or are born small for gestational age. Similarly, to what we observed in our cohort [6,20].

Two thirds of untreated asymptomatic newborns would develop CS symptoms in the first 3 to 8 weeks of life and almost all at 3 months of life [4]. In our cohort, 78% of patients were asymptomatic at birth, but 83% of these (i.e. 66% of the total cohort) developed clinical CS manifestations within the first 2 month, while the rest of the asymptomatic patients presented some symptoms within 21 months. The widespread and profound consequences of delayed CS diagnoses due to inadequate serological evaluation of pregnancies and newborns are unacceptable, and should prompt a re-evaluation of the health system, and the chain of missed opportunities and oversights that leads to these cases.

In addition, the analysis of the clinical manifestations demonstrated a high morbidity. Bone affectation was observed in a great number of patients. Bone involvement (particularly osteochondritis) revealed by radiological tests and often limb pain, was the most frequent finding in our study. This proved to be a useful sign of active congenital syphilis in asymptomatic newborns, as suggested by others authors [4,21]. Hepatosplenomegaly has also been reported as one of the most common findings [22]. In our study, both hepatosplenomegaly and dermatological manifestations followed in frequency after bone lesions. The high rate of bone involvement observed could may be inherent to the age of our cohort because we are a referral pediatric center without a maternity ward.

Regarding laboratory findings, hepatitis, jaundice and anemia are well-known manifestations of CS [6]. Although 54% of our patients presented with hepatomegaly, only 36% presented with alterations in liver enzymes. In contrast, anemia was present in a high percentage of patients, requiring blood transfusion in 5/61 cases. Among the related causes of liver injury, the direct action of TPA causing hepatocellular cholestasis without anatomical bile ducts alteration has been described [23]. https://paperpile.com/c/ueTD2m/rUsJx.

In order to evaluate CNS involvement, CSF testing (cell count, protein, and VDRL) has been recommended since neurosyphilis is the most concerning manifestation of CS [6,24]. It was suggested that asymptomatic newborn lumbar puncture examination be avoided [1,25]. In our study, only 10% (5/50) of patients for whom CSF was available showed alterations. However, there are many issues with diagnostic accuracy, for example: white-cell count and protein values vary with age; VDRL/RPR antibodies can passively transfer into the CSF, and *Treponema pallidum* was reported detected with normal CSF results [25].

The need for CSF evaluation has been questioned, due to its lack of accuracy in confirming or discarding CNS involvement [6,26]. In our study, spinal lumbar puncture resulted in frequent blood contamination, low VDRL sensitivity and did not change the therapeutic management. Also, no differences in clinical after-treatment evolution was observed between patients with or without CSF evaluation. In our opinion, the value of lumbar punctures in this population is very limited, does not provide reliable data as a prognostic marker and does not justify the risks and suffering associated with the routine use of this diagnostic procedure.

We observed that 89% of asymptomatic patients recovered after treatment with intravenous aqueous crystalline penicillin G. Although 14.7% showed Jarisch-Herxheimer reaction, these events were self-limiting. In the retrospective analysis, we detected one patient that, in the hospitalization and under the suspicion of sepsis, received intravenous ceftriaxone for 10 days with favorable clinical evolution. The possibility of using alternative drugs to penicillin for the treatment of CS is a field that needs to be explored in depth, particularly given the worldwide shortage of penicillin [27,28,29].

Reactive serological data in newborns and infants can be confusing due to passive placental transfer of maternal antibodies and often result in unnecessary hospitalization and treatments [10,30]. Non-treponemal tests (VDRL/RPR) titers 4 times higher than those of the mother are criteria for surveillance case and, higher concentration of VDRL/RPR antibodies is related to the activity of the infection [8,18,30]. In our study, 4-fold higher RPR titers were absent in a

considerable proportion of infected patients and did not help to reach CS diagnosis while a sustained decrease in antibody titers was observed as an indicator of therapeutic response. Similarly, the role of treponemal (TPHA, FTA-Abs) test in the diagnosis of CS was questioned [30]. In our study TPHA titers remained reactive after treatment so this test was not useful for treatment monitoring.

New techniques for CS diagnosis are needed, since a long term follow-up is required to confirm the diagnosis, especially in asymptomatic infants [1]. Nucleic acid- based amplification assays, such as polymerase chain reaction (PCR) have greatly improved sensitivity and specificity of direct detection of TPA in vitro. However, there is sparse information regarding use of PCR in clinical settings. A clinical study is under development in our center (Clinical trials Identifier: NCT04084379) in order to validate the sensitivity and specificity of this test for CS diagnosis.

Low patient adherence to follow up was observed. Although this attrition is multifactorial in nature, socioeconomic vulnerability, such as low-income and high internal migration, constituted a barrier for health care access in our cohort.

Like other developing countries in our region, the high prevalence of syphilis required that serial syphilis testing should be done to women (during pregnancy and the perinatal period) and to newborn born without any prenatal care [7,11,17,18]. This is a barrier to medical follow up and treatment and local reports have identified inequalities in differentiating areas of unmet demand and the access to health care centers [31].

In a recent report, the PAHO pointed out that there is room to improve in every healthcare process in Latin America to help eliminate CS [9]. Although Argentina has reported favorable changes in increasing reports of maternal and congenital cases, continuing difficulties persist, such as: the lack of qualified staff; the unclear definitions of responsibilities for maternal cases active follow-up; the need to scale up access to timely diagnosis and treatment and the lack of CS case audits.

Many issues make CS a great challenge for the pediatrician as is shown in this study. A high number of symptomatic infants with a completely preventable and curable fetal infection were observed, resulting from inadequate detection and treatment of syphilis during pregnancy. It is mandatory to improve the health care of syphilis, both in terms of screening and treatment, during pregnancy.

Strict enforcement of existing guidelines, and strengthening of the health care system, particularly in the area of primary care and prevention during pregnancy, should go a long way towards eliminating this severe, but completely preventable disease. Furthermore, we noticed a failure to appropriately screen pregnant patients. Without a reassessment of the practices and prejudices of the entire health system, which allow these women to fall through the cracks of the health systems, tied with strong enforcement of existing guidelines (and clear consequences for those who do not follow them), CS cases will continue to take place.

As conclusion, our data revealed that it is still common for patients to arrive at a diagnosis when expressing early or late CS. In our opinion, the value of lumbar punctures is very limited and does not justify the risks and suffering associated to the routine use of this diagnostic procedure. Each CS that was diagnosed reflected a fault [12] in spite of health programs and the first and second level of health care level granted. However, these patients could still reach an early diagnosis and treatment if an adequate maternal and infant serology review is performed in the pediatric practice.

As a particular contribution of this study, the commented issues indicate the quality of our data that respond to a need in our region, and exemplified the real world obstacles to reach the EMTC of syphilis in our media. In addition to the future actions described by PAHO, we believe that policies that monitor the population accessibly to care and the different levels of

articulation of health care will be extremely helpful in the immediate epidemiological landscape.

## Supporting information

**S1 Fig. Flowchart of patient inclusion process and complementary studies**
(TIF)

**S2 Fig. Symptomatic newborn.** A) Term male newborn, adequate for gestational age with widespread erythroderma, vesiculobullous eruptions and desquamation (pemphigus syphiliti-cus). (B) Bone involvement lesions in the same patients are shown (arrow).
(TIF)

**S1 Text. STROBE checklist.**
(DOC)

## Acknowledgments

The authors wish to thank native speaker Lesley Speakman for revising the English of the manuscript.

## Author Contributions

**Conceptualization:** Samanta Moroni, Facundo García Bournissen.

**Data curation:** Alejandra Destito Solján, Samanta Moroni, Nicolás Gonzalez, Guillermo Moscatelli.

**Formal analysis:** Alejandra Destito Solján, Nicolas Falk.

**Funding acquisition:** Jaime M. Altcheh.

**Investigation:** Alejandra Destito Solján, Nicolás Gonzalez, Guillermo Moscatelli, Facundo García Bournissen.

**Methodology:** Nicolas Falk, Griselda Ballering.

**Project administration:** Luciana Noemí Garcia, Jaime M. Altcheh.

**Resources:** Griselda Ballering, Jaime M. Altcheh.

**Software:** Nicolas Falk.

**Supervision:** Jaime M. Altcheh.

**Validation:** Samanta Moroni, Facundo García Bournissen.

**Visualization:** Luciana Noemí Garcia.

**Writing – original draft:** Luciana Noemí Garcia, Alejandra Destito Solján.

**Writing – review & editing:** Samanta Moroni, Jaime M. Altcheh.

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
