## [Decision Letter · Decision Letter 0]

12 Aug 2020

Dear Dr García,

Thank you very much for submitting your manuscript "Congenital syphilis in Argentina: experience in a pediatric hospital" for consideration at PLOS Neglected Tropical Diseases. As with all papers reviewed by the journal, your manuscript was reviewed by members of the editorial board and by several independent reviewers. In light of the reviews (below this email), we would like to invite the resubmission of a significantly-revised version that takes into account the reviewers' comments. 

We cannot make any decision about publication until we have seen the revised manuscript and your response to the reviewers' comments. Your revised manuscript is also likely to be sent to reviewers for further evaluation.

Sincerely,

Peter J. Krause

Deputy Editor

Richard Phillips

Deputy Editor

Reviewer's Responses to Questions

**Key Review Criteria Required for Acceptance?**

**Methods**

-Are the objectives of the study clearly articulated with a clear testable hypothesis stated?

-Is the study design appropriate to address the stated objectives?

-Is the population clearly described and appropriate for the hypothesis being tested?

-Is the sample size sufficient to ensure adequate power to address the hypothesis being tested?

-Were correct statistical analysis used to support conclusions?

-Are there concerns about ethical or regulatory requirements being met?

Reviewer #1: (No Response)

Reviewer #2: The main objective of the study is to describe a Buenos Aires (Argentina) Children's Hospital congenital syphilis experience over a period of 32 years (1987-2019). To do so the study team conducted a retrospective medical record review of all patients diagnosed with presumed congenital syphilis over a 32 year time period .

In this reviewer's opinion, this study lacks scientific rigor and attention to detail. The study lacks clarity in the case definition of congenital syphilis. This is a major flaw of this study and in this reviewer's opinion it makes the unsuitable for publication in its current format.

**Results**

-Does the analysis presented match the analysis plan?

-Are the results clearly and completely presented?

-Are the figures (Tables, Images) of sufficient quality for clarity?

Reviewer #1: (No Response)

Reviewer #2: This manuscript is poorly written. It lacks coherence and it is truly impossible to read without getting confused and distracted. Tables and fixtures are of poor quality and denote a lack of attention to detail, which is not simply a lack of command of the English language.

**Conclusions**

-Are the conclusions supported by the data presented?

-Are the limitations of analysis clearly described?

-Do the authors discuss how these data can be helpful to advance our understanding of the topic under study?

-Is public health relevance addressed?

Reviewer #1: (No Response)

Reviewer #2: This paper is a hastily completed description of CS cases over a 30 year period. The author's did not attempt at correlating their findings to historical data, epidemiological trends in Argentina or Latin America, or the many papers that have been published over the past decade describing congenital syphilis throughout the world. Based on the manuscript it appears that the author's made no attempt at describing how they ascertained cases and if they believe their cases are a true estimate of all cases seen at their hospital.

**Editorial and Data Presentation Modifications?**

Reviewer #1: (No Response)

Reviewer #2: This paper needs major modifications and revisions to make it suitable for publication in any journal. While a review of congenital syphilis in any country in Latin America is worth while, it must be carefully completed and must add new information about the disease or a new perspective for how a review such as this one adds new information and helps define strategies for control of the disease. 

The methods section needs to be revised and strengthen and carefully described. Tables and figures need to be revised and aligned with existing models for how to describe congenital syphilis.

**Summary and General Comments**

Reviewer #1: Authors compiled a large case series of congenital cases in Argentina from 1987 through 2019. The clinical description, follow ups and treatments are very well described and of clinical interest for the medical community. Few minor revisions can improve the main message of the manuscript

- Abstract: Please adjust to mention the sequelae at follow ups. This is perhaps one of the most relevant findings of the study

- Results: Please expand on the clinical findings of; "In 4 patients persistent proteinuria was observed as a sequelae

207 of renal involvement and in 1 patient knee arthrosis occurred due to bone damage". Was renal failure observed? what age were those 4 patients? was the proteinuria attribuited to congenital syphilis?

- Discussion: Please expand on reasons why only 47% of pregnant women completed an adequate syphilis screening? What public health measures can be implemented to link those pregnant women to care? Is it routine to screen newborns and infants for Congenital syphilis born who were born without any prenatal care? 

- Discussion: As a reader, I'd like to hear about more actionable recommendations in your community to prevent additional cases of congenital syphilis.

Reviewer #2: In summary this study is not suitable for publication in its current format.

PLOS authors have the option to publish the peer review history of their article (what does this mean?). If published, this will include your full peer review and any attached files.

Reviewer #1: Yes: Andres Henao

Reviewer #2: No
---

## [Decision Letter · Decision Letter 1]

25 Nov 2020

Dear Dr García,

We are pleased to inform you that your manuscript 'Congenital syphilis in Argentina: experience in a pediatric hospital' has been provisionally accepted for publication in PLOS Neglected Tropical Diseases.

Best regards,

Peter J. Krause

Deputy Editor

Richard Phillips

Deputy Editor

Reviewer's Responses to Questions

**Key Review Criteria Required for Acceptance?**

**Methods**

-Are the objectives of the study clearly articulated with a clear testable hypothesis stated?

-Is the study design appropriate to address the stated objectives?

-Is the population clearly described and appropriate for the hypothesis being tested?

-Is the sample size sufficient to ensure adequate power to address the hypothesis being tested?

-Were correct statistical analysis used to support conclusions?

-Are there concerns about ethical or regulatory requirements being met?

Reviewer #1: (No Response)

**Results**

-Does the analysis presented match the analysis plan?

-Are the results clearly and completely presented?

-Are the figures (Tables, Images) of sufficient quality for clarity?

Reviewer #1: (No Response)

**Conclusions**

-Are the conclusions supported by the data presented?

-Are the limitations of analysis clearly described?

-Do the authors discuss how these data can be helpful to advance our understanding of the topic under study?

-Is public health relevance addressed?

Reviewer #1: (No Response)

**Editorial and Data Presentation Modifications?**

Reviewer #1: (No Response)

**Summary and General Comments**

Reviewer #1: (No Response)

PLOS authors have the option to publish the peer review history of their article (what does this mean?). If published, this will include your full peer review and any attached files.

Reviewer #1: **Yes: **Andres F. Henao-Martinez

---

## [Editor Report · Acceptance letter]

30 Dec 2020

Dear Dr García,

We are delighted to inform you that your manuscript, "Congenital syphilis in Argentina: experience in a pediatric hospital," has been formally accepted for publication in PLOS Neglected Tropical Diseases.

Best regards,

Shaden Kamhawi

co-Editor-in-Chief

Paul Brindley

co-Editor-in-Chief
